# Study of the Effects of Current Imbalance in a Multiphase Buck Converter for Electric Vehicles

**Iván Alfonso Reyes-Portillo** [1,*], **Abraham Claudio-Sanchéz** [2], **Jorge Alberto Morales-Saldaña** [1], **Jesús Darío Mina-Antonio** [2], **Edgardo Marvel Netzahuatl-Huerta** [1], **Luisana Claudio-Pachecano** [3], **Mario Ponce-Silva** [2] **and Ericka Reyes-Sánchez** [1]

1   Engineering Electronic Department, Universidad Autonoma de San Luis Potosi—CIEP,
San Luis Potosi 78290, Mexico; jmorales@uaslp.mx (J.A.M.-S.); A325084@alumnos.uaslp.mx (E.M.N.-H.);
ericka.sanchez@uaslp.mx (E.S.-R.)
2   Engineering Electronic Department, Tecnologico Nacional de Mexico—CENIDET, Cuernavaca 62490, Mexico;
abraham.cs@cenidet.tecnm.mx (A.C.-S.); jmina@cenidet.edu.mx (J.D.M.-A.);
mario.ps@cenidet.tecnm.mx (M.P.-S.)
3   Engineering Mechanical Department, Tecnologico Nacional de Mexico—CENIDET,
Cuernavaca 62490, Mexico; luisana.cp@cenidet.tecnm.mx
*   Correspondence: A318057@alumnos.uaslp.mx; Tel.: +52-294-121-45-77

**Abstract:** The excessive use of fossil fuels has caused great concern due to modern environmental problems, particularly air pollution. The above situation demands that different areas of research aim at a sustainable movement to reduce $CO_2$ emissions caused by non-renewable organic fuels. A solution to this problem is the use of Electric Vehicles (EV) for mass transportation of people. However, these systems require high-power DC/DC converters capable of handling high current levels and should feature high efficiencies to charge their batteries. For this application, a single-stage converter is not viable for these applications due to the high current stress in a switch, the low power density, and its low efficiency due to higher switching losses. One solution to this problem is Multiphase Converters, which offer high efficiency, high power density, and low current ripple on the battery side. However, these characteristics are affected by the current imbalance in the phases. This paper is focused on the study of the effects of the current imbalance in a Multiphase Buck Converter, used as an intermediate cover between a power supply and the battery of an EV. Analyzing the efficiency and thermal stress parameters in different scenarios of current balance and current imbalance in each phase.

**Keywords:** battery charger; DC/DC; Multiphase Buck Converter; current balance

## 1. Introduction

Global warming, climate change, and the destruction of the ozone layer are factors that are forcing governments to look for feasible solutions to reduce the use of fossil fuels [1,2]. Automotive transportation is a major source of pollution, due to a large number of vehicles on the road every day around the world. For this reason, the progressive change from vehicles powered by fossil fuels to electric vehicles, as sustainable mobility, is essential for helping to solve the environmental problems mentioned above [3]. Although the majority of electrical energy is indeed generated by non-renewable energy sources, the trends are being directed towards the generation of energy with renewable energy sources (photovoltaic panels, wind generators, fuel cells) [4]. Electric vehicles are a solution that promises to reduce the damage to the environment, due to their sustainable characteristics described in [5,6]. Another aspect is the reduction of noise pollution because the electrical engine noise is almost imperceptible [7,8]. In this way, emissions and noise pollution levels are reduced in the places where these vehicles circulate, thus reducing respiratory diseases. Other advantages of the electric vehicle are related to the efficient use of the energy consumed by

the electric motor, which is 90% [9]. Unlike the internal combustion engine whose energy efficiency percentage is between 20% and 25%, that is, only that percentage of thermal energy is transformed into mechanical energy [10]. The absence of a gearbox contributes to better acceleration response and excellent kinematic behavior. Moreover, electric vehicles implement regenerative braking systems recovering energy during this process, a characteristic that conventional vehicles do not have, which produce losses in the form of heat dissipated due to friction [11,12]. However, there are still many challenges to be addressed in these complex electrical systems [13]. A clear example is the charging process of electric vehicle batteries, whose voltage is 48V, and requires high current levels to achieve a fast and complete state of charge [14–17]. The characteristics of rechargeable batteries have always been critical in the electric vehicle development [18–21]. To carry out a fast and adequate process of battery charge, a high power density DC/DC converter is required [2,22,23]. The converters for EV´s battery charging applications must have high power density with low ripple of current and voltage, especially on the battery side. In addition, the converter has to meet basic industry requirements, such as high efficiency, low cost, and compact component size [24]. Several circuit designs for high power applications have been published [25,26]. Most of these designs require large coupled inductors and high voltage, high current devices [27]. The size of the components are large, which decreases the power density [28]. A multiphase converter is a good solution for high power, high current, and low ripple of voltage and current output applications [29]. The advantages of multiphase techniques are: reducing the stress on the devices, reducing the filter size, and decreasing the voltage and current ripple at the converter output [30,31]. However, most published papers require current sensors in the control loops of each phase to achieve balanced phase currents and improve the dynamic response of the converter [32–34]. The current imbalance depends mainly on the duty cycle, the inductance, and the parasitic resistances in each phase that integrate the converter [34,35]. The current imbalance is one of the most important problems for multiphase converters in practical applications. Several approaches have been proposed to solve this problem, including current sensor-based [35–37] and non-sensor-based methods [38,39]. In [38], the current imbalance is caused by variations in the parasitic resistances of each phase. Most papers are concerned with correcting the current balance in multiphase converters but do not focus on the study of the physical effects of this current imbalance on the converter. This paper presents a study of the effects of the current imbalance in the Multiphase Buck Converter (MBC). To analyze the disadvantages of the operating converter with current imbalance, an average modelwith parasitic resistances is proposed to obtain the parameters involved in the current imbalance phenomenon in each phase. In addition, the physical effects of the converter in current balance and imbalance are experimentally compared. Figure 1 shows the schematic of the four-phase MBC for electric vehicle battery charging. The EV battery charger considers an MBC, which has been designed with four phases, where ($V_G$) is the input voltage and $V_o$ is the output voltage. The output current is formed by the four phases of the converter, making this structure ideal for fast battery charging. Finally, this paper is organized as follows: Section 2 describes the converter operation and shows the equations for sizing of the passive elements. In Section 3, the average model and the effects of parasitic resistors on the DC current balance are presented. Section 4 presents the effects of parasitic resistances in the converter. Section 5 presents the experimental results and, finally, Section 6 presents the conclusions and recommendations.

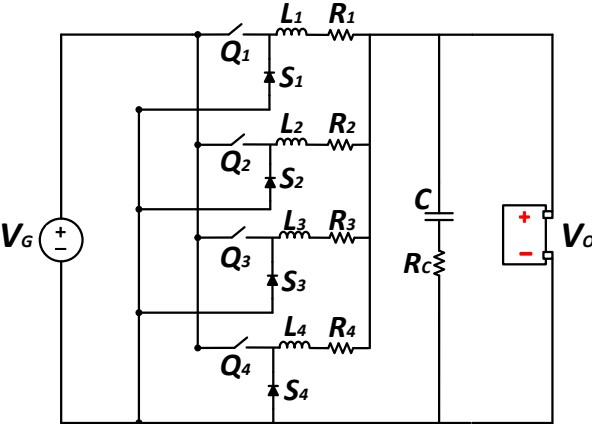

**Figure 1.** MBC consisting of four phases for battery charging in EVs.

## 2. Operation of the Multiphase Buck Converter

The converter presents two modes of conduction: Continuous Conduction Mode (CCM) where the inductor current never reaches zero, and Discontinuous Conduction Mode (DCM) where the inductor current is zero during an interval of time in this paper; only the CCM is analyzed. The MBC is proposed for EV battery charging applications operating with overlapping control signals. The equivalent circuits proposed for the "ON and OFF" operating modes are only valid for the following duty cycle operating range $0.25 < D < 0.5$. The CCM has different operating states during a switching period ($T_s$), four in "ON mode" which correspond to turning ON switches $Q_1$, $Q_2$, $Q_3$ and $Q_4$ with 90° delay between each phase, while the diodes remain ON except for the diode where switch Q is active. Figure 2 presents the operating states in "ON mode" considering the duty cycle signals ($D$) of each phase in a switching period.

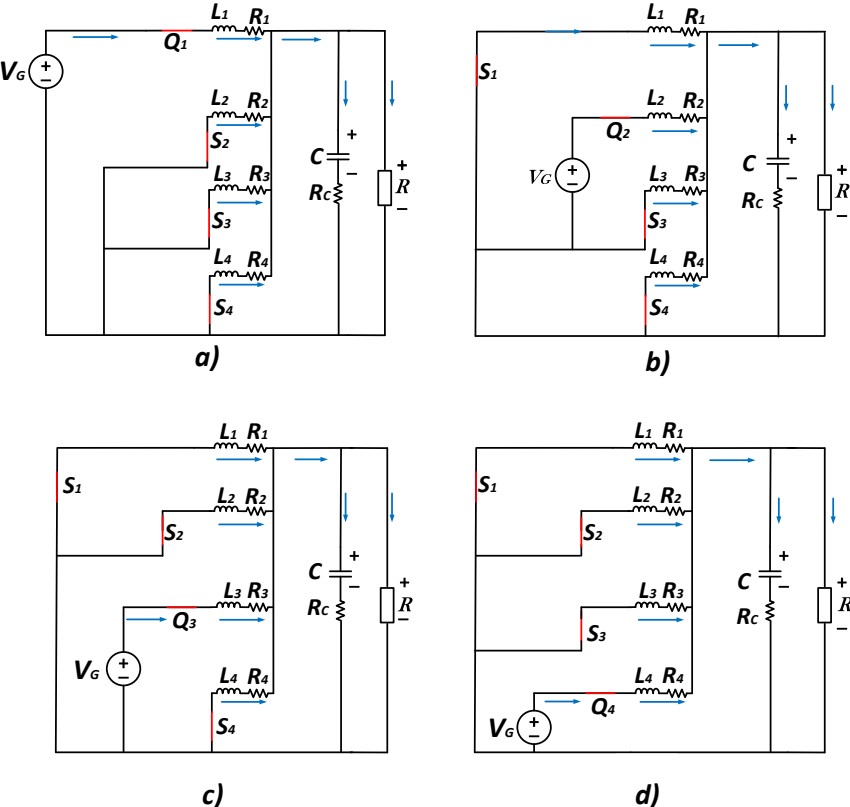

**Figure 2.** During the battery charging process, the equivalent circuits in "ON mode" for each phase elements, (**a**) $D_1t_2$, (**b**) $D_2t_4$, (**c**) $D_3t_6$, (**d**) $D_4t_8$.

During OFF mode, there are overlaps in the control signals, during a lapse of the switching period. The switch $Q_1$ overlaps with the next phase that is 90° away; this phenomenon occurs for each phase in turn. Phase four overlaps with the next switching period. With the overlap of the control signals, the source ($V_G$) is always present in the eight operating states of the converter. Figure 3 shows the four equivalent circuits of the OFF mode.

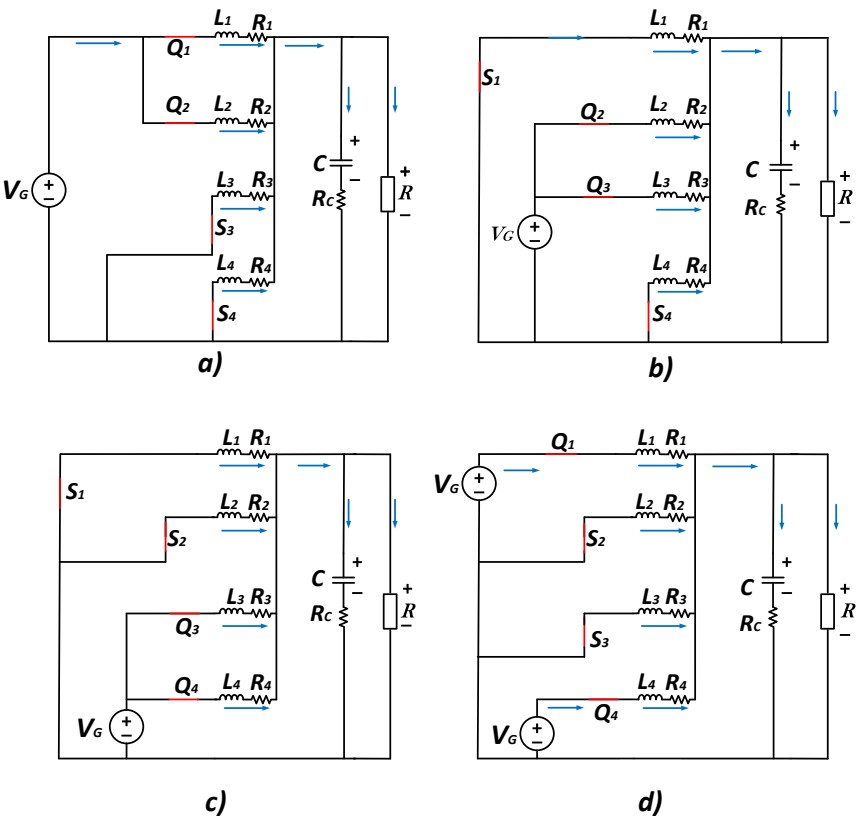

**Figure 3.** During the battery charging process, the equivalent circuits in "OFF mode" for each phase elements, (**a**) $D_1D_2t_3$; (**b**) $D_2D_3t_5$; (**c**) $D_3D_4t_7$; (**d**) $D_4D_1t_1$.

Figure 4 shows the waveforms, during one switching period, for the steady-state analysis of the MBC.

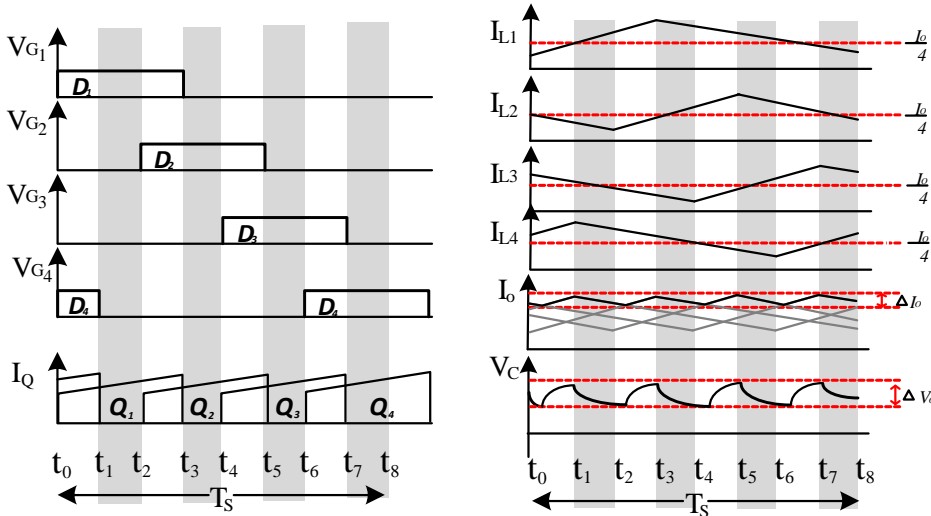

**Figure 4.** Waveforms during a commutation period.

Figure 4 shows that the inductor currents at the output of the MBC are interleaved, due to the effect of the phase difference between the control signals. The same improves the behavior of the converter by reducing the output current ripple and the converter output voltage. It also increases the frequency of the voltage and current at the output. It is important to note that reducing the current ripple size at the converter output also helps in decreasing the capacitance value at the output [40–42]. Another advantage of the MBC is that the pulsed current demanded from the source ($V_G$) is four times lower compared to a conventional buck converter, considering the same power requirements. The advantage of operating the converter with overlapping control signals is that the discontinuity of the input current demanded from the $V_G$ source is eliminated. By analyzing the steady-state signals (Figure 4), the steady-state (1)–(6) of the MBC was determined:

$$I_{L_1} = \frac{D_1 V_G}{4R} \tag{1}$$

$$I_{L_2} = \frac{D_2 V_G}{4R} \tag{2}$$

$$I_{L_3} = \frac{D_3 V_G}{4R} \tag{3}$$

$$I_{L_4} = \frac{D_4 V_G}{4R} \tag{4}$$

$$V_C = D_1 V_G \tag{5}$$

To obtain an effective voltage transformation ratio for CCM, the volt–second balance of the voltage signal in one of the inductors is applied, considering that the duty cycle (D) is the same in each phase. The voltage transformation ratio, also known as the gain, is given by:

$$M = \frac{Vo}{V_G} = D \tag{6}$$

The following expressions establish the value of the inductance that the converter needs in each phase to operate in its steady state. It is important to consider the percentage of the current ripples ($\Delta_{I_L}$) according to the load:

$$L_1 = \frac{V_o(1 - D_1)}{\Delta I_{L_1} f_s} \tag{7}$$

$$L_2 = \frac{V_o(1 - D_2)}{\Delta I_{L_2} f_s} \tag{8}$$

$$L_3 = \frac{V_o(1 - D_3)}{\Delta I_{L_3} f_s} \tag{9}$$

$$L_4 = \frac{V_o(1 - D_4)}{\Delta I_{L_4} f_s} \tag{10}$$

To determine the value of the capacitor, it is necessary to consider the converter output frequency ($f_{S_{out}}$) given by:

$$f_{S_{out}} = 4fs \tag{11}$$

The capacitor value is determined with the next expression:

$$C = \frac{V_o(1D_1)D_1}{8\Delta V_C L_1 f_{S_{out}}^2} \tag{12}$$

In (12), the output frequency is four times higher than the switching frequency ($f_s$) as a result of the ripple cancellation effect [30]. Consequently, the output capacitor reduces its capacitance value considerably.

## 3. Modeling of the Converter

The modeling of the converter was carried out through the state space technique, analyzing the different configurations defined by the operation modes presented in Figures 2 and 3. This modeling considers the parasitic resistances in the electronic elements. To simplify the analysis, the following considerations were made:

$$R_1 = R_{Q_1} + R_{L_1} + R_{track} \tag{13}$$

$$R_2 = R_{Q_2} + R_{L_2} + R_{track} \tag{14}$$

$$R_3 = R_{Q_3} + R_{L_3} + R_{track} \tag{15}$$

$$R_4 = R_{Q_4} + R_{L_4} + R_{track} \tag{16}$$

$$R_C = R_{ESR} + R_{track} \tag{17}$$

where $R_{Q_{1,2,3,4}}$ are the resistances $R_{DS_{on}}$ present in the switches, $R_{L_{1,2,3,4}}$ are the resistances present in the inductors and $R_{track}$ is the resistance of the copper track, present in each phase. State variables are chosen as: $x_1 = I_{L_1}$, $x_2 = I_{L_2}$, $x_3 = I_{L_3}$, $x_4 = I_{L_4}$ and $x_5 = V_C$. The state variables correspond to the currents in the inductors and the voltage of the output capacitor. Using the technique of averaged states, it is possible to obtain a unified average model, given by:

$$\dot{x}_1 = -\frac{R_1 + R_C}{L_1}x_1 - \frac{R_C}{L_1}x_2 - \frac{R_C}{L_1}x_3 - \frac{R_C}{L_1}x_4 - \frac{1}{L_1}x_5 + \frac{D_1}{L_1}V_G + \frac{R_C}{L_1}I_o \tag{18}$$

$$\dot{x}_2 = -\frac{R_C}{L_2}x_1 - \frac{R_2 + R_C}{L_2}x_2 - \frac{R_C}{L_2}x_3 - \frac{R_C}{L_2}x_4 - \frac{1}{L_2}x_5 + \frac{D_2}{L_2}V_G + \frac{R_C}{L_2}I_o \tag{19}$$

$$\dot{x}_3 = -\frac{R_C}{L_3}x_1 - \frac{R_C}{L_2}x_2 - \frac{R_3 + R_C}{L_3}x_3 - \frac{R_C}{L_3}x_4 - \frac{1}{L_3}x_5 + \frac{D_3}{L_3}V_G + \frac{R_C}{L_3}I_o \tag{20}$$

$$\dot{x}_4 = -\frac{R_C}{L_4}x_1 - \frac{R_C}{L_3}x_2 - \frac{R_C}{L_4}x_3 - \frac{R_4 + R_C}{L_4}x_4 - \frac{1}{L_4}x_5 + \frac{D_4}{L_4}V_G + \frac{R_C}{L_4}I_o \tag{21}$$

$$\dot{x}_5 = -\frac{1}{C}x_1 + \frac{1}{C}x_2 + \frac{1}{C}x_3 + \frac{1}{C}x_4 - \frac{1}{C}I_o \tag{22}$$

and in the form $\dot{x} = Ax + Bu$, we obtain the average model matrix

$$
\begin{bmatrix} \dot{x}_1 \\ \dot{x}_2 \\ \dot{x}_3 \\ \dot{x}_4 \\ \dot{x}_5 \end{bmatrix} =
\begin{bmatrix}
-\frac{R_1+R_C}{L_1} & -\frac{R_C}{L_1} & -\frac{R_C}{L_1} & -\frac{R_C}{L_1} & -\frac{1}{L_1} \\
-\frac{R_C}{L_2} & -\frac{R_2+R_C}{L_2} & -\frac{R_C}{L_2} & -\frac{R_C}{L_2} & -\frac{1}{L_2} \\
-\frac{R_C}{L_3} & -\frac{R_C}{L_3} & -\frac{R_3+R_C}{L_3} & -\frac{R_C}{L_3} & -\frac{1}{L_3} \\
-\frac{R_C}{L_4} & -\frac{R_C}{L_4} & -\frac{R_C}{L_4} & -\frac{R_4+R_C}{L_4} & -\frac{1}{L_4} \\
\frac{1}{C} & \frac{1}{C} & \frac{1}{C} & \frac{1}{C} & 0
\end{bmatrix}
\begin{bmatrix} x_1 \\ x_2 \\ x_3 \\ x_4 \\ x_5 \end{bmatrix} +
\begin{bmatrix}
\frac{D_1}{L_1} & \frac{R_C}{L_1} \\
\frac{D_2}{L_2} & \frac{R_C}{L_2} \\
\frac{D_3}{L_3} & \frac{R_C}{L_3} \\
\frac{D_4}{L_4} & \frac{R_C}{L_4} \\
0 & -\frac{1}{C}
\end{bmatrix}
\begin{bmatrix} V_G \\ I_o \end{bmatrix}
$$

Solving for the DC values of the state variables, we employ expression (23)

$$X = -A^{-1}BU \tag{23}$$

where $U$ is the input voltage ($V_G$) and the output current ($I_o$). The equations of the currents in each phase of the converter and the output voltage capacitor are given by:

$$I_{L_1} = \frac{V_G[R_3R_4(D_1D_2) + R_2R_4(D_1D_3) + R_2R_3(D_1D_4)] + I_o(R_2R_3R_4)}{R_1R_2R_3 + R_1R_2R_4 + R_1R_3R_4 + R_2R_3R_4} \tag{24}$$

$$I_{L_2} = \frac{V_G[R_3R_4(D_2D_1) + R_1R_4(D_2D_3) + R_1R_3(D_2D_4)] + I_o(R_1R_3R_4)}{R_1R_2R_3 + R_1R_2R_4 + R_1R_3R_4 + R_2R_3R_4} \tag{25}$$

$$I_{L_3} = \frac{V_G[R_2R_4(D_3D_1) + R_1R_4(D_3D_2) + R_1R_2(D_3D_4)] + I_o(R_1R_2R_4)}{R_1R_2R_3 + R_1R_2R_4 + R_1R_3R_4 + R_2R_3R_4} \tag{26}$$

$$I_{L_4} = \frac{V_G[R_2R_3(D_4 - D_1) + R_1R_4(D_4 - D_2) + R_1R_2(D_4 - D_3)] + I_o(R_1R_2R_3)}{R_1R_2R_3 + R_1R_2R_4 + R_1R_3R_4 + R_2R_3R_4} \tag{27}$$

$$V_C = \frac{V_G[D_1R_2R_3R_4 + D_2R_1R_3R_4 + D_3R_1R_2R_4 + D_4R_1R_2R_3] - I_o(R_1R_2R_3R_4)}{R_1R_2R_3 + R_1R_2R_4 + R_1R_3R_4 + R_2R_3R_4} \tag{28}$$

Equations (24)–(28) contain the parameters involved in the effect of current imbalance, describing that the MBC converter is susceptible to current imbalance due to factors such as small variations of the duty cycle (D) between each phase and variations of the values of the parasitic resistances present in the converter.

*Current Balance Control Strategy*

The paper presents a hysteresis controller that integrates the functions of voltage regulation, current balance, and the 90° phase shift of the control signals for the MBC. The implementation of these functions is done employing analog circuits taking into account simplicity, low cost, and functionality as main objectives. The controller architecture is shown in Figure 5 in the form of a block diagram. The scheme was implemented in a commercial controller [43]. The voltage regulation function only requires sensing of the output voltage $V_o$ and, for the current balance function, it requires sensing of the voltage at the chopper output between the switches Q and S. It is not necessary to sense the current directly in each phase, which reduces the controller complexity and implementation cost. The input voltage ($V_G$) of the MBC is 120 V, the reference voltage ($V_{ref}$) is 3 V, and the output voltage ($V_o$) was considered to be 48 V. With these considerations, the current balance scheme was implemented in the commercial controller [43]. This paper focuses on the effects of current imbalance on the MBC converter and not on the controller performance.

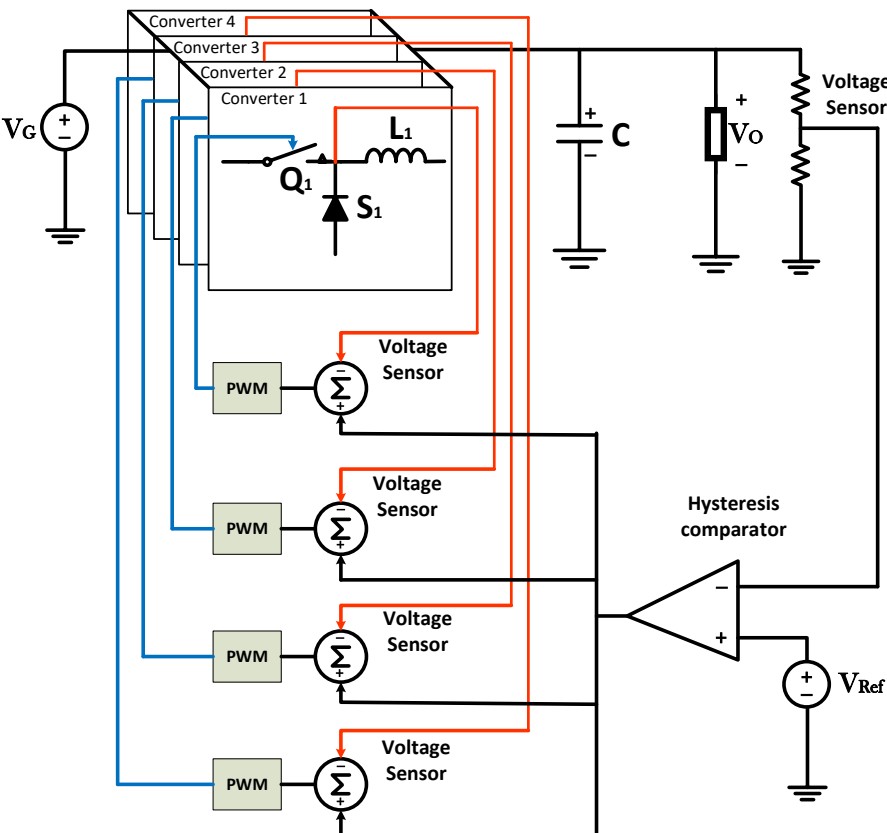

**Figure 5.** General control diagram for current balance by a phase based on [43].

## 4. Parasitic Resistance Effect

One of the main causes of current imbalance in the different phases is the variation of the parasitic resistances present in power semiconductor devices and other elements (L and C) [35,44]. This paper only considers the effect of parasitic resistance variations, as this parameter is more susceptible to variations due to external effects, such as temperature, copper track length, and component manufacturing. The typical variations in the values of parasitic resistances in inductors and capacitors reported by suppliers are ±2%. These variations increase in value, due to temperature effects as shown in Figure 6. The temperatures recommended by the manufacturers at which the switches should operate are between 20~80 °C. Figure 6 shows the behavior of $R_{DS_{on}}$ concerning the temperature present in the switches, where the red frame indicates the temperatures recommended by the manufacturers and their interaction with the variation of the resistance $R_{DS_{on}}$.

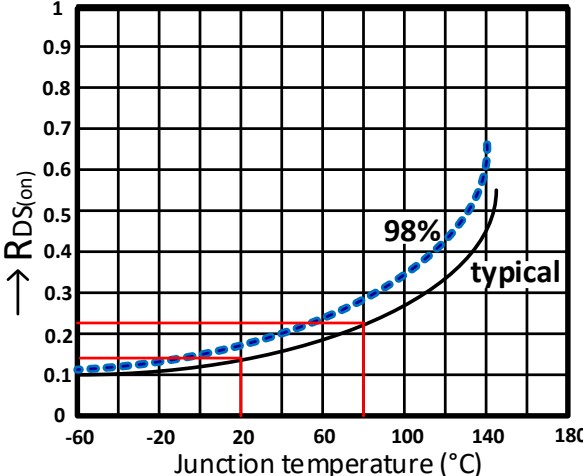

**Figure 6.** Variation of the $R_{DS_{ON}}$ with respect to temperature variation.

The typical variation of $R_{DS_{on}}$ at the same temperature, considering the fabrication linked to each particular switch is 2%. These variations increase in value when the power switches operate at different temperatures. This effect is observed in the curves described in Figure 6. The data of these curves are provided by the manufacturer of the switches. The variations of the parasitic resistances in the MBC directly affect the current balance [31]. This effect is described by expressions (24)–(27). To analyze the current imbalance due to the effect of parasitic elements, an MBC was designed considering an input voltage $(V_G)$ of 120 V and an output current $(I_o)$ of 40 A. Table 1 shows the list of parameters and devices used to perform the current imbalance analysis.

**Table 1.** Design parameters.

| Symbol | Parameter | Value |
|--------|-----------|-------|
| $V_G$ | Input Voltage | 120 V |
| $P_o$ | Output Power | 1920 W |
| $V_o$ | Output voltage | 48 V |
| $I_o$ | Output current | 40 A |
| $f_s$ | Frequency | 103 kHz |
| $L_1, L_2, L_3, L_4$ | Inductors | 150 uH |
| $C$ | Capacitor | 2 uF |
| $D$ | Duty cycle per phase | 40% |
| $Q_1, Q_2, Q_3, Q_4$ | MOSFETs | C20N60CFD |
| $S_1, S_2, S_3, S_4$ | Diodes | TO-220-L |
| $R_1, R_2, R_3, R_4$ | Parasitic resistance | 130 mΩ |
| $R_C$ | Capacitor ESR | 30 mΩ |

It should be noted that an output voltage of 48 V was selected based on current trends for battery voltage in electric vehicles presented in [14–17]. Current imbalance analysis was simulated in PSpice. To emulate these effects, parasitic resistance variations and an open-loop operation were considered to establish critical scenarios. Table 2 shows the values of the variations of the parasitic resistors due to the effect of temperature.

**Table 2.** Percentage of parasitic variations per phase.

| Case | Phase 1-$R_1$ | | Phase 2-$R_2$ | | Phase 3-$R_3$ | | Phase 4-$R_4$ | |
|------|------|------|------|------|------|------|------|------|
| | % | mΩ | % | mΩ | % | mΩ | % | mΩ |
| I | +10 | 143 | +10 | 143 | +10 | 143 | 0 | 130 |
| II | +10 | 143 | +10 | 143 | 0 | 130 | 0 | 130 |
| III | +10 | 143 | +10 | 143 | 0 | 130 | −10 | 117 |
| IV | 0 | 130 | +10 | 143 | −10 | 117 | −10 | 117 |

Figure 7 presents the current waveforms in the MBC, considering the different cases established in Table 2. In the MBC simulation, all duty cycles (D) of the same value were considered.

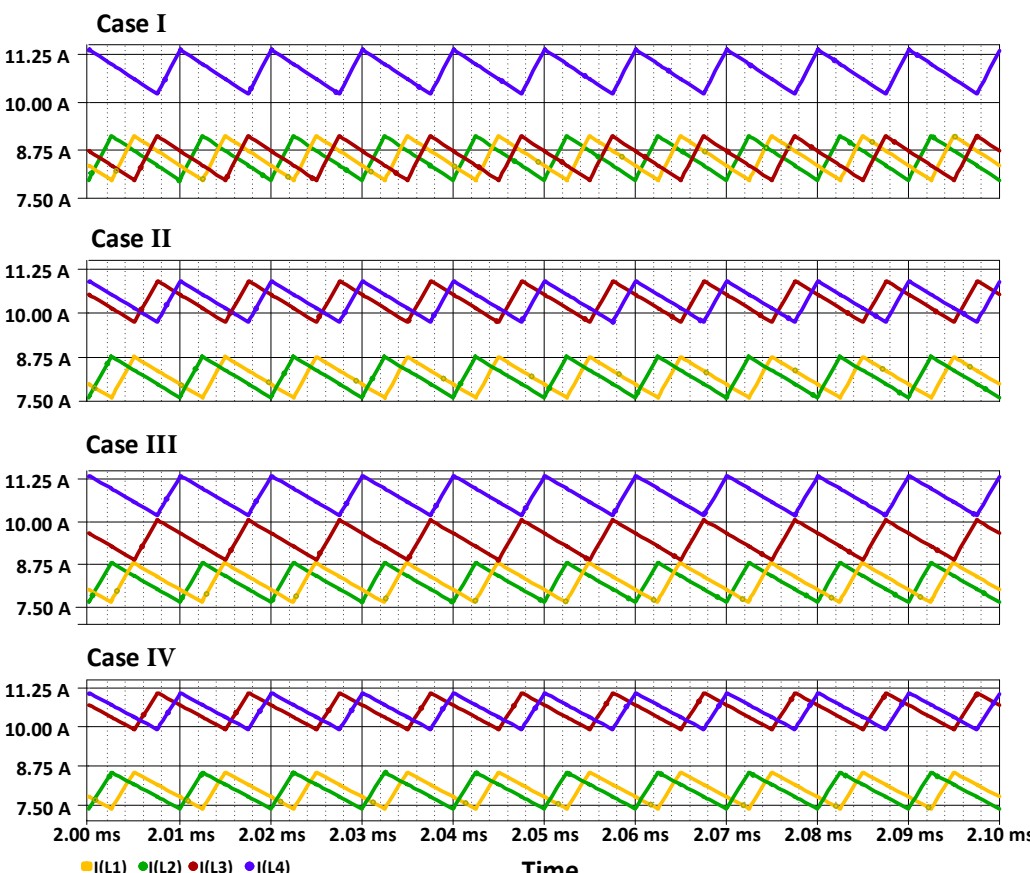

**Figure 7.** Four cases of current imbalance with respect to Table 2.

- Case I: In phase four, a current of 10.7 A flows, this is due to the lower value of parasitic resistance. Having the highest current in this phase, the switching losses and thermal stress will be higher, while the other phases maintain a balanced current flow with a value of 8.7 A.
- Case II: In phases three and four, the current flow is 10.7 A; this is because the parasitic resistors have less opposition to the passage of current in these phases, while in phases one and two, the current value is 8.2 A.

- Case III: In phase four, it demands a current of 10.7 A, phase three demands 9.5 A, while phases one and two only demand 8.2 A.
- Case IV: In phases three and four, a current of 10.7 A is demanded, in phase one a current of 9.5 A is demanded, and, in phase two, the current is 8.2 A.

## 5. Experimental Analysis

A functional prototype of the MBC was built (Figure 8), to analyze the different current imbalance scenarios according to the variables established in Table 2. The current imbalance tests were carried out in an open-loop, where the converter operates in the scenarios used for the simulation.

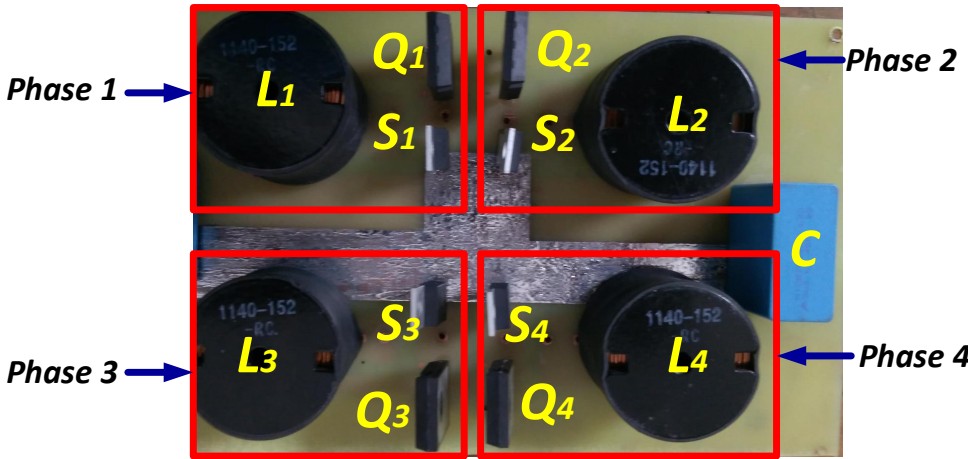

**Figure 8.** Functional prototype of the MBC.

Figure 9 shows the control signals used in the switches. The control signals have an offset of 90° between each phase, a frequency of 103 kHz, and a duty cycle (*D*) of 40%.

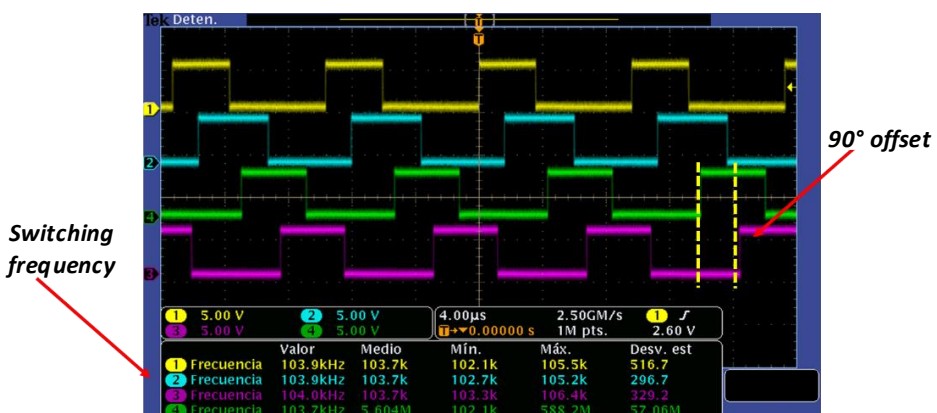

**Figure 9.** Commutation signals for phases with the corresponding 90º phase shift and duty cycle (40%).

The functional prototype was tested with different current demands at the output, maintaining a constant duty cycle. Table 3 shows the current values per phase in the MBC when operating in an open-loop and with current imbalance. Figure 10 presents the current imbalance cases described in Table 3. Due to the complexity of varying the parasitic resistances of the components, the external parameter $R_{track}$ of the copper tracks was used to consider the different current imbalance scenarios.

**Table 3.** Experimental cases of current imbalance per phase.

| Case | Phase 1 ($A$) | Phase 2 ($A$) | Phase 3 ($A$) | Phase 4 ($A$) |
|------|---------------|---------------|---------------|---------------|
| I    | 7.02          | 6.61          | 6.37          | 2.99          |
| II   | 8.83          | 8.39          | 3.47          | 3.86          |
| III  | 7.06          | 6.68          | 5.13          | 3.21          |
| IV   | 13.1          | 3.78          | 2.58          | 8.29          |

The cases presented in Table 3 are described below:

- Case I: In Phase one, it transfers 7.02 A, being the phase with the highest current. Phases two and three present similar current values, with a value of 6.61 A and 6.37 A, while phase three is the one with the lowest current value transferred to the output. In this case, the output current ($I_o$) was 23 A.
- Case II: Phases one and two present more switching losses and high levels of thermal stress; this is due to the fact that they are the branches that transfer more current to the output, while phases three and four present values close to 3.6 A. In this case, the output current ($I_o$) was 25.55 A.
- Case III: Phase one transfers 7.02 A, in phase two, a current of 6.68 A circulates and phase three sends 5.13 A to the converter output, while phase four is where less current circulates. In this case, the output current ($I_o$) was 22.08 A.
- Case IV: This case presents the most critical case of current imbalance, phase one transfers 13.1 A to the output of the converter, phase four 8.29 A and phase two 3.78 A, while only 2.58 A circulates through phase three. In this case, the output current ($I_o$) was 27.75 A.

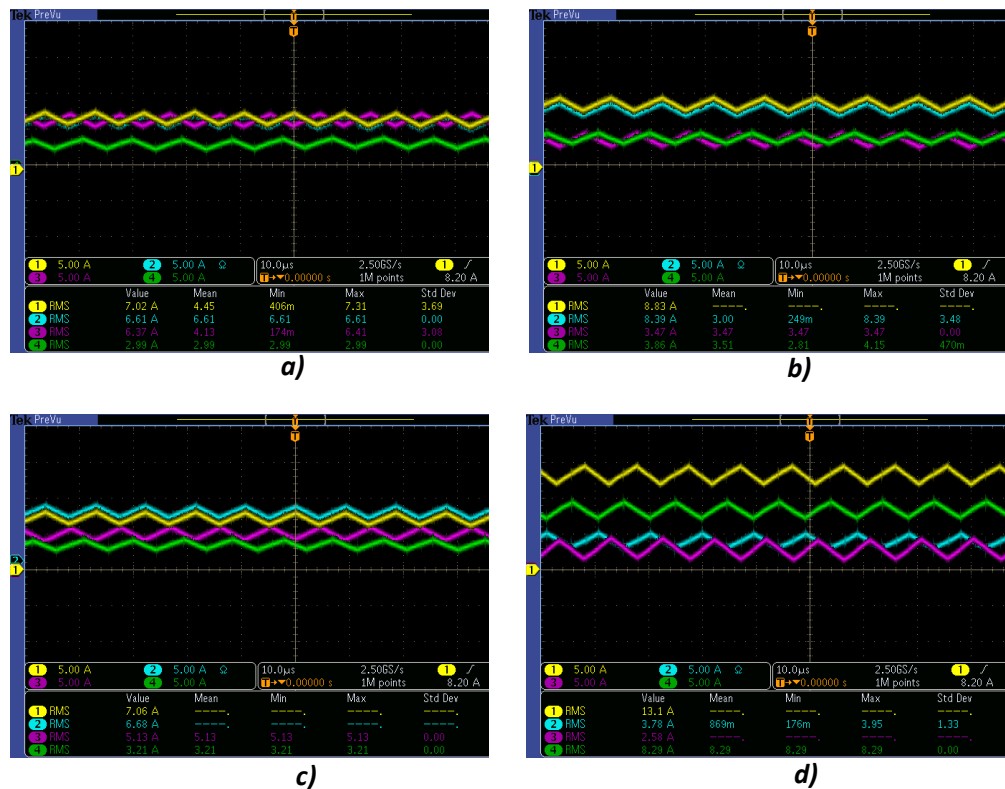

**Figure 10.** Current imbalance in the MBC prototype. (**a**) Case I, (**b**) Case II, (**c**) Case III, (**d**) Case IV.

The experimental tests of the previous cases were performed in a controlled environment. The initial ambient temperature for each test was set at 22 °C. In each test, it was corroborated that all semiconductor devices started at the aforementioned ambient

temperature. To carry out the current imbalance tests, the semiconductor devices were oversized for the protection of the experimental prototype. In order to reduce the effects of current imbalance, it is of utmost importance to have a scheme that guarantees the current balance in each phase during the battery charging process in an electric vehicle.

Figure 11 shows the operation of the control scheme based on Figure 5 to balance the current flow per phase. It is observed that the system does not manage to equalize the currents in all phases; however, there is a great improvement with respect to the critical cases presented in Figure 10. Figure 11a shows the balanced currents, where phase one transfers 8.30 A, being the branch where the highest current flows, while, in phase three, 6.52 A circulates, being the phase that transfers less current to the output. The other two phases operate with current values close to 7.4 A. In this test, the value of the output current is 30 A.

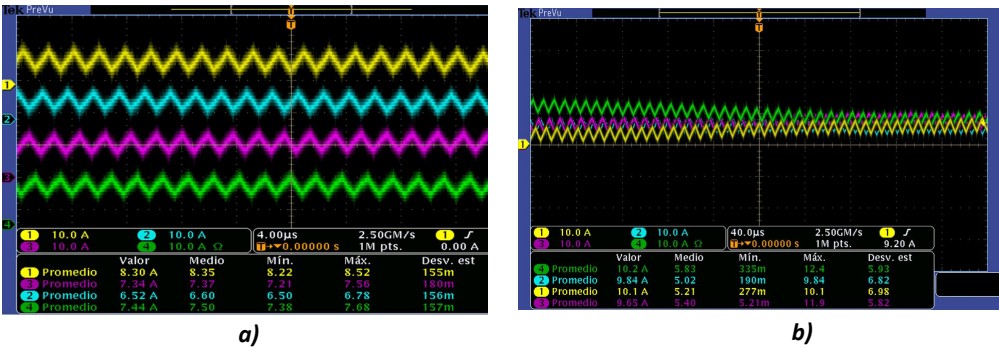

**Figure 11.** Current signals with the current balance scheme (**a**) balanced currents, (**b**) operation of the current balance scheme ($\approx 240\,\mu S$ stabilization response).

The theoretical balanced current in each phase for the case of Figure 11a is 7.5 A per phase. The current balance for phase one varies by +15%, phase two by −2%, phase three has a variation of −13%, and phase four has an imbalance of −1%. In Figure 11b, the converter was subjected to a current demand of $10A$ per phase, considering a theoretical balance. In phase one, the variation between the theoretical balance and the experimental balance was +2%, in phase two, the variation was −1.6%, in phase three, the variation was +1% and for phase four the variation was −3.5%. Figure 11b shows how the control scheme forces the currents to balance in each phase. This test was subjected to the nominal output current of 40 A. It is important to note that the study focuses on the physical effects suffered by the MBC converter during critical cases of current imbalance. However, to have comparative data of the results obtained with balanced currents scenarios, it was necessary to have a control scheme that offers these characteristics. However, the objective of this work is not to evaluate the performance of the control scheme.

*Thermal Stress Analysis of the Converter*

The current imbalance directly influences the temperatures of the switches. The phase in which the higher current flows suffers higher thermal stress, which damages the switching devices and causes more frequent system failures. Figure 12 shows the temperature spectrum of the switches in a current imbalance scenario. In this test, it is observed that the highest thermal stress is found in the phase two and phase four transistor.

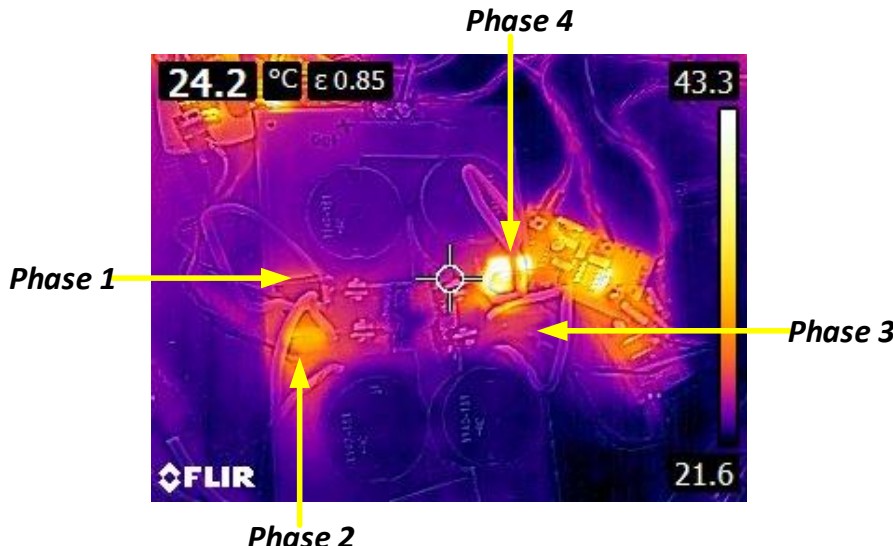

**Figure 12.** Temperature spectrum with imbalanced phases of the converter.

The current balance in the phases benefits from an equal distribution of thermal stress in the switches. Figure 13 presents the temperature spectrum during a test of the MBC operating with the current balance scheme.

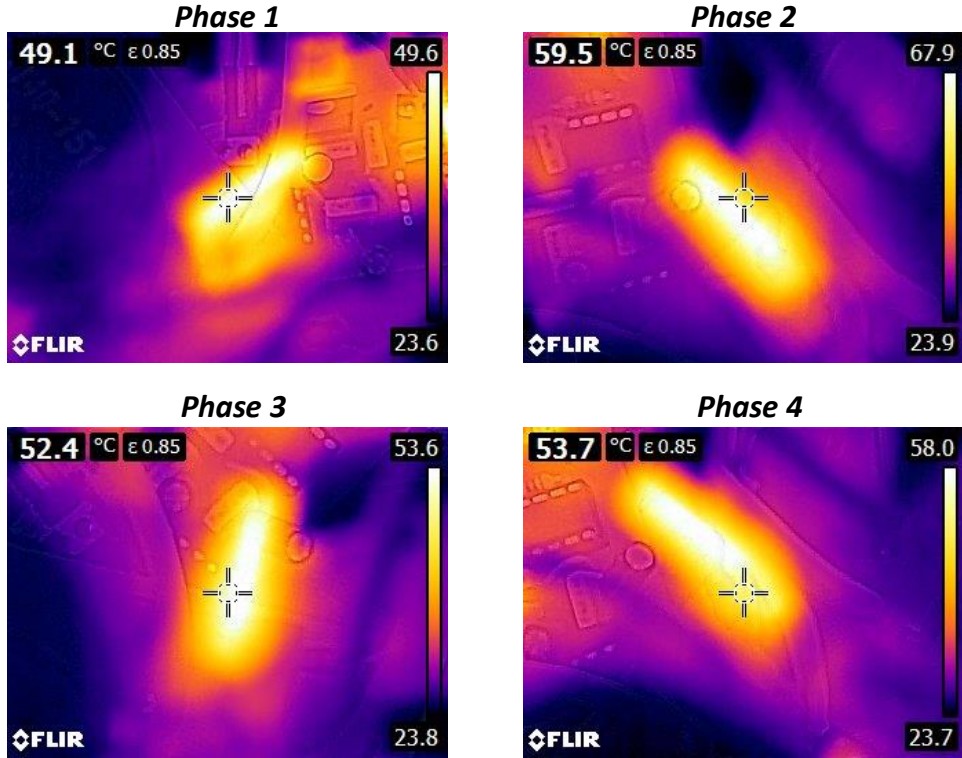

**Figure 13.** Temperature spectrum with balanced phases of the converter.

The temperature variations shown by the switches in Figure 13 are between 49.1 °C and 59.5 °C, being a suitable temperature to operate the manufacturer recommendations. Figure 14 shows the temperature curves for different power scenarios. The data obtained are with and without the current balance scheme.

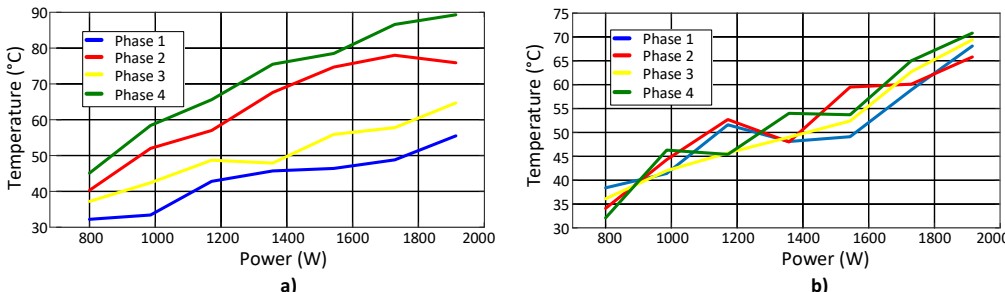

**Figure 14.** Behavior of temperature in switches (**a**) imbalanced currents, (**b**) balanced currents.

Figure 14a shows the behavior of the temperature of the switches during the current imbalance, after 60 s of operation in a controlled environment of 22 °C as the initial temperature. An indirect way to measure the current imbalance per phase is by employing the temperature of the semiconductors. On the other hand, Figure 14b shows the temperature behavior of the switches in each phase with the balanced current. The temperatures of the transistors in each phase present insignificant variations compared to the temperatures presented without the current compensation scheme per phase. The efficiency is directly affected by the balance of currents in each phase, due to thermal stress and higher dissipated power losses in switches. Figure 15 shows the comparison of the efficiency curves in the MBC with balanced and imbalanced currents. The nominal power of the converter is 1.9 kW. The converter achieved an efficiency of 91.87% with the current balance scheme operating at its nominal power, while, with imbalanced currents, the efficiency was 89.41%.

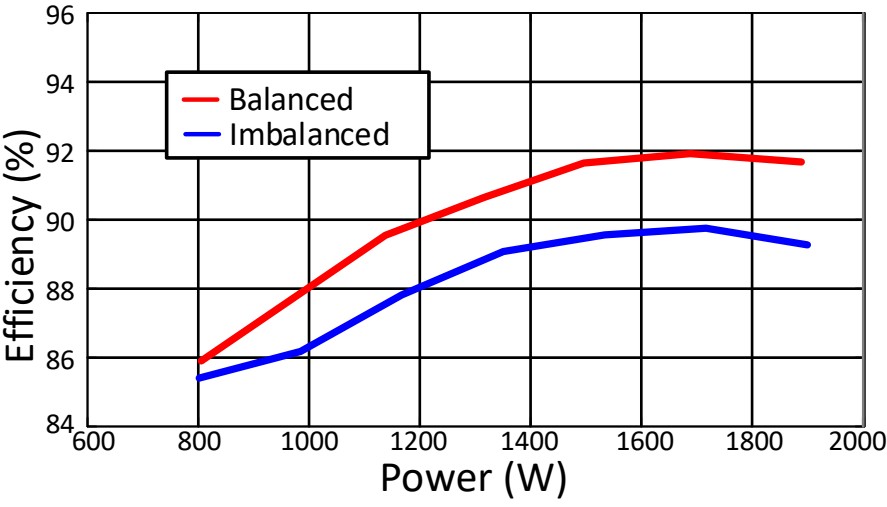

**Figure 15.** Converter efficiency as a function of nominal power.

## 6. Conclusions

This paper presented the study of the effects of current imbalance in the MBC during the battery charging process. It focused mainly on the comparative study of the physical phenomena suffered by the converter, such as thermal stress and efficiency. It was proved that the effect of current imbalance directly affects the favorable characteristics of the MBC, such as power density and efficiency. To corroborate the theoretical results, an experimental prototype of the converter with a nominal power of 1.9 kW was built and tests were carried out under different power scenarios. The behavior of the switches temperatures in different power scenarios was presented experimentally, considering the balanced and imbalanced currents. The efficiency curves of the converter with and without the current balance scheme were presented, showing that current imbalance has a negative influence on efficiency. The efficiency at nominal power of the MBC shows a difference of 2.46% between the balanced and imbalanced currents. This converter has interesting and favorable characteristics for battery

charging in electric vehicles due to its high power density and simple dynamic characteristics, but it is important to consider a good current balancing scheme in order not to lose these favorable characteristics. Future work includes the study of the effects of current imbalance on the battery life cycle and the analysis of the different current balancing strategies considering aspects such as cost and performance in the MBC.

**Author Contributions:** Conceptualization, I.A.R.-P., A.C.-S.; validation, J.D.M.-A.; formal analysis; L.C.-P., E.R.-S.; investigation, J.A.M.-S., M.P.-S., E.M.N.-H. All authors have read and agreed to the published version of the manuscript.

**Funding:** This research received no external funding.

**Institutional Review Board Statement:** Not applicable.

**Informed Consent Statement:** Not applicable.

**Data Availability Statement:** Study did not report any data.

**Acknowledgments:** The authors would like to thank the Tecnologico Nacional de Mexico—CENIDET and the Universidad Autonoma de San Luis—CIEP for the material resources provided.

**Conflicts of Interest:** The authors declare no conflict of interest.

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
