# Peer review of "Study of the Effects of Current Imbalance in a Multiphase Buck Converter for Electric Vehicles"

_wevj, doi:10.3390/wevj13050088_

Round 1
Reviewer 1 Report
Congratulations to the authors for a very nice and useful work.
Author Response
Muchas gracias por sus comentarios y observaciones sobre el trabajo “Estudio de los Efectos del Desequilibrio de Corriente en un Convertidor Buck Multifásico para Vehículos Eléctricos”.

Reviewer 2 Report
The paper presents the study of a Multiphase Buck Converter, as an intermediate stage between the power source and the battery of an EV, emphasizing not only the effects that parasitic DC resistances have in the current imbalance between phases, by means of the analysis of a dynamic model of the converter but also in how this imbalance affects the efficiency and the thermal stress of the converter. The topic is timely and well researched however, I have the following suggestions to enhance the overall quality of the paper. Please, add more new references related to the topic in the introduction part, please highlight the novelty of the paper in comparison with other similar work, add a flowchart to explain various processes of the research for better readability, the paper has some typos and grammar mistakes, so please get help from an English editor if applicable or check the whole writing, rewrite the abstract and conclusion for more research focus, and please add a complete section about the future work. Thank you so much.Author Response
Thank you very much for your comments and observations on the paper "Study of the effects of current imbalance in a multiphase Buck converter for electric vehicles".

Reviewer 3 Report
The proposed paper titled "Study of the Effects of Current Imbalance in a Multiphase Buck Converter for Electric Vehicles" presents a study of imbalance currents in multiphase DC/DC converter. The study shows how imbalanced DC resistances in each phase might lead to imbalance currents, and shows the effect of imbalance currents on the efficiency and on the thermal stress of the converter.
Here are my comments concerning the paper:
-While the introduction introduces well the context and the reason for using a multi-level DC/DC converter for electric vehicles, it does not discuss about existing analysis of the current balance problem, and does not discuss of literature solutions to this problem. For instance, there already exists many current balancing methods that could be cited.
-The introduction should explain how the proposed study brings something new compared to the existing literature. The authors should clarify what is the novelty of their analysis compared to previous works in the field.
-"Due to the parasitic elements as well as the small variations to the duty ratio (D), considering that D is equal in all phases." I think that this sentence should be rewritten.
-What is the purpose of deriving equations (24)-(28). Are they used in the design of the current balance control strategy? I think that this should be further explained.
-How has figure 6 been obtained? Does it come from a reference? If it does, the reference should be indicated in the figure caption.
-In the experimental analysis, how were the four experimental cases of current unbalanced phases implemented?
-Is Figure 11 obtained with the control scheme shown in Fig.5? If this the case, this should be detailed.
Author Response
Thank you very much for your comments and observations on the paper "Study of the effects of current imbalance in a multiphase Buck converter for electric vehicles".

Reviewer 4 Report
This paper presents many critical issues, and its implementation should be improved. The authors spent many parts of the text focusing on general environmental issues but do not provide sufficient details about the novelty and the motivations behind the work. The text of the paper presents many too long and complex sentences and in many points, it is difficult to follow.
Specific comments in the following:
In the Abstract:
Please specify where this Multiphase Buck converter is used. According to the treated levels of currents, I think the authors had in mind to deal with chargers, but this is not made clear in the text. This should be clearly stated. Is the discussed structure devoted to offboard chargers, onboard chargers, both cases, EV electronic loads in general?
The sentence is too long and complex, please split it and make it simpler. “This paper presents the study of a Multiphase Buck Converter, as an intermediate stage between the power source and the battery of an EV, emphasizing not only in the effects that parasitic DC resistances have in the current imbalance between phases, by means of the analysis of a dynamic model of the converter, but also in how this imbalance affects the efficiency and the thermal stress of the converter.”
In the Introduction:
Please take care of spelling mistakes: Example “sucha as” in the introduction.
Too many repetitions are present. For example, the two sentences: “The electric vehicles are a solution that promises to reduce the damage to the environment, due to their sustainable characteristics described in [4,5]. Electric vehicles respond to current ecological demands in various aspects, such as the principal demand is the reduction of CO2 emissions into the environment, improving air quality.” gives the same meaning.
Please specify the percentage of energy efficiency consumed by an electric motor for the comparison with IC engines.
It should be better to explain, even in short, how the absence of a gearbox can improve the kinematic behavior of an electric motor compared to IC engines.
There is no link between the discussion of the power train and the charging process of a battery. Moreover, the charging process doesn’t come under electromechanical issues.
The authors mentioned a “standard voltage” of 48V. Generally, the battery voltage in hybrid or battery EVs ranges from 200-800V. Please better explain this aspect also in relation to the voltage associated with renewable energy.
Introduction fails to provide the advantages of buck converter, multiphase buck converter, why there are current imbalances (causes are provided in section 3, but in the introduction, an overview is required), and how these unbalance currents make an impact on battery. Better provide relevant reference papers on current imbalances on MBC.
State of the art is missing. Many DC-DC converters exist in the literature for EV battery charging applications. Why MBC is considered?
Section 2:
What are overlapping control signals? Please refer to them in the control signals figure.
It is not clear what the authors are trying to explain regarding ON and OFF modes.
Figure 3a says D1D2t3, where Q1 is OFF but the equivalent circuit shows Q1 is ON. Isn’t should be only D2t3?
In the sentence “voltage transformation ratio in MCC”, what is MCC?
Again, too many repetitions. E.g. the “one of the advantages of this converter is the decrease in the output current ripple ( ΔIo )” contains information already mentioned.
Section 3:
Define all equation terms from 13-17. For example, RQ1, RL1, Rtrack.
A proper explanation is required for equations 18 to 22. Are they defined for both ON and OFF states of switches?
At equilibrium, in equation 23, u should be capitalized.
In the whole text, the authors do not make clear what is the cause of the current unbalance?
In section 3.1:
figure 5 is not clear. What the red lines denote? The figure reports a current sensor, but only the voltage between Q and S is mentioned in the text. What is the output of the hysteresis comparator? Please specify alla these aspects.
Again, repetition regarding “The scheme was implemented in a commercial controller [19].”
Section 4:
“Figure 5 shows the behavior of RDS.” It is figure 6, not figure 5.
Explain figure 6, mention what are red and black lines refer to?
Unbalance currents in phases are caused by the variation in parasitic resistance and other elements (first sentence in section 4). But, in case 1, phase 4 has zero variation in parasitic resistance. So isn’t the phase 4 current a balanced current compared to other phase currents? The authors say in case 1 other phases maintain a balanced current flow with a value of 8.7A. How this is justified?
A brief and clear explanation is needed for the Pspice simulation results.
Section 5:
Please specify the theoretical (Pspice simulation) results are from the open-loop or closed loop. Because these are compared with experimental results of both open loop and closed loop cases. Please justify how theoretical results can be compared with experimental results in both cases.
“Figure 13 b) shows how the control scheme forces the currents to balance in each phase.” It is figure 11 b, please correct it.
I suggest to the authors to specify why such design parameters mentioned in table 1 are considered? The battery rated voltage is around 200 to 800V (nominal voltage of each cell is 3.2 / 4.2 V) in EVs (If Li-ion batteries are considered). So, why 48V as an output?
Section 5.1:
What is the greatest current? Change it to huge or large. Same for thermal stress.
“In this test it is observed that the greatest thermal stress is found in the transistor of phase two and phase four.” The context was not clear in which “cases” the authors are referring to these phases 2 and 4.
It is not clear what the authors want to prove from the closed-loop experimental results. In a closed-loop, the current imbalances are reduced, which is an obvious situation. If possible, I suggest authors to focus on current imbalances that occur during closed-loop operation.
In on-field usages, the converter will operate in closed-loop control. So, what is the point in comparing the closed-loop results with the open-loop?
Moreover, only parasitic resistance variation is considered. The other element’s variation is not mentioned in this paper. I suggest authors to better explain these things in the literature.
Author Response

(The authors gave the same response as above.)

Round 2
Reviewer 3 Report
The authors have addressed most of my comments. In my opinion, the new version of the manuscript is suitable for publication.
Author Response

(The authors gave the same response as above.)

Reviewer 4 Report
The authors have modified the paper considering the comments of the previous turn of revision. However, two minor things still need to be changed/Included.
- In section 3, equations from 13 to 17: Resistance Rtrack is not defined anywhere so it should be clearly introduced and defined.
- In section 4 first paragraph, the Figure indication is wrong: “Figure 5 shows the behavior of RDSon concerning the temperature present in the switches
Generally, I suggest to improve the quality of the figures 6, 14, 15 and all tables.
Author Response

(The authors gave the same response as above.)
